# IL-33 Reduces Saturated Fatty Acid Accumulation in Mouse Atherosclerotic Foci

**DOI:** 10.3390/nu16081195

**Published:** 2024-04-17

**Authors:** Yukako Hosomi, Takuro Okamura, Kimiko Sakai, Hiroki Yuge, Takashi Yoshimura, Saori Majima, Hiroshi Okada, Takafumi Senmaru, Emi Ushigome, Naoko Nakanishi, Takashi Satoh, Shizuo Akira, Masahide Hamaguchi, Michiaki Fukui

**Affiliations:** 1Department of Endocrinology and Metabolism, Graduate School of Medical Science, Kyoto Prefectural University of Medicine, Kyoto 602-8566, Japan; hy0226@koto.kpu-m.ac.jp (Y.H.); d04sm012@koto.kpu-m.ac.jp (T.O.); k-sakai@koto.kpu-m.ac.jp (K.S.); yuge@koto.kpu-m.ac.jp (H.Y.); yoshimu3@koto.kpu-m.ac.jp (T.Y.); saori-m@koto.kpu-m.ac.jp (S.M.); conti@koto.kpu-m.ac.jp (H.O.); semmarut@koto.kpu-m.ac.jp (T.S.); emis@koto.kpu-m.ac.jp (E.U.); naoko-n@koto.kpu-m.ac.jp (N.N.); michiaki@koto.kpu-m.ac.jp (M.F.); 2Department of Immune Regulation, Graduate School and Faculty of Medicine, Tokyo Medical and Dental University (TMDU), Tokyo 113-8510, Japan; secr.mbch@tmd.ac.jp; 3Department of Host Defense, Research Institute for Microbial Diseases (RIMD), Osaka University, Suita 565-0871, Japan; a-office@biken.osaka-u.ac.jp; 4Laboratory of Host Defense, World Premier Institute Immunology Frontier Research Center, Osaka University, Suita 565-0871, Japan

**Keywords:** interleukin-33, atherosclerosis, saturated fatty acids, type 2 innate lymphocytes, atherosclerotic foci

## Abstract

The cellular and molecular mechanisms of atherosclerosis are still unclear. Type 2 innate lymphocytes (ILC2) exhibit anti-inflammatory properties and protect against atherosclerosis. This study aimed to elucidate the pathogenesis of atherosclerosis development using atherosclerosis model mice (ApoE KO mice) and mice deficient in IL-33 receptor ST2 (ApoEST2 DKO mice). Sixteen-week-old male ApoE KO and ApoEST2 DKO mice were subjected to an 8-week regimen of a high-fat, high-sucrose diet. Atherosclerotic foci were assessed histologically at the aortic valve ring. Chronic inflammation was assessed using flow cytometry and real-time polymerase chain reaction. In addition, saturated fatty acids (palmitic acid) and IL-33 were administered to human aortic endothelial cells (HAECs) to assess fatty acid metabolism. ApoEST2 DKO mice with attenuated ILC2 had significantly worse atherosclerosis than ApoE KO mice. The levels of saturated fatty acids, including palmitic acid, were significantly elevated in the arteries and serum of ApoEST2 DKO mice. Furthermore, on treating HAECs with saturated fatty acids with or without IL-33, the Oil Red O staining area significantly decreased in the IL-33-treated group compared to that in the non-treated group. IL-33 potentially prevented the accumulation of saturated fatty acids within atherosclerotic foci.

## 1. Introduction

Atherosclerosis is a persistent inflammatory condition [1]. Dysfunction of the endothelium within the arterial wall causes monocytes to adhere, take up lipoproteins, differentiate, and accumulate into foamy macrophages, which gradually develop into atherosclerotic plaques [2]. Atherosclerosis is caused by unstable plaque disruption, platelet aggregation, and thrombosis, leading to the narrowing and occlusion of blood vessels and the development of cardiovascular disease [3].

Lymphocytes critical in innate immunity and devoid of antigen receptors are called innate lymphocytes (ILCs) and are associated with lifestyle-related diseases [4,5,6]. ILCs differentiate from progenitor cells common to the lymphoid lineage and, like T lymphocytes, produce cytokines but lack antigen receptors. ILCs are categorized into ILC1, ILC2, and ILC3 subunits. ILC1 secretes interferon (IFN)-γ and exhibits antimicrobial activity against intracellular microorganisms. ILC2 produces interleukin (IL)-5, IL-9, and IL-13, contributing to defense against parasite defense and allergic diseases. ILC3 generates IL-22 in response to IL-23 and IL-1β, participating in innate immunity against fungi and extracellular pathogens [7].

IL-33 is a member of the IL-1 family, which includes IL-1β and IL-18, and has potent immunomodulatory functions [8]. IL-33 is localized in the nucleus of vascular endothelial and epithelial cells. Upon injury, IL-33 is rapidly translocated extracellularly, stimulating basophils, mast cells, and ILC2 to produce Th2 cytokines (involved in innate-type allergies), or stimulates Th2 with antigens, enhancing IL-5 and IL-13 production (involved in acquired-type allergies). Thus, IL-33 participates in both innate- and acquired-type allergies. IL-33 receptors are present in Th2 cells, regulatory T cells, natural killer (NK) cells, ILC2, macrophages, eosinophils, basophils, and mast cells. Increased IL-33 activity affects the onset and severity of allergic diseases; however, ILC2 has anti-inflammatory effects and protects against atherosclerosis [9]. T cells infiltrate the vascular wall and respond to the presence of oxidized low-density lipoprotein (ox-LDL) with smooth muscle and endothelial cells, producing cytokines and inflammatory mediators, which, depending on their phenotype, can promote plaque formation (Th1) or inhibit inflammatory changes (Th2) [10].

Unlike IL-1β and IL-18, which primarily promote Th1-related responses, IL-33 induces the production of Th2 cytokines, such as IL-5 and IL-13, and increases serum immunoglobulin levels. A previous study has shown that the exogenous administration of IL-33 to ApoEKO mice induced Th1-Th2 production in vivo and reduced ox-LDL antibody levels. This report also reported that exogenous administration of IL-33 to ApoEKO mice markedly increased levels of IL-4, -5, and -13, decreased levels of IFNγ in serum and lymph node cells, and decreased the atherosclerotic lesion area in the aortic sinus [11]. Moreover, IL-33 promotes the browning of adipose tissue, prevents obesity, and decreases ILC2 levels in the adipose tissue of obese mice [12]. IL-33 may show a protective role in atherosclerotic lesions [13]. M1 macrophages remove ox-LDL via scavenger receptors to form foam cells and undergo apoptosis. In contrast, M2 macrophages phagocytose apoptotic foam cells and exhibit anti-inflammatory properties [14]. However, the cellular and molecular mechanisms of atherosclerotic disease remain unclear. Therefore, this study aimed to elucidate the mechanisms underlying changes in innate immunity and the development of chronic inflammation in a mouse model of atherosclerosis (ApoEKO mice) and mice lacking the IL-33 receptor, ST2 (ApoEST2DKO mice).

## 2. Materials and Methods

### 2.1. Animals

All animal experimental protocols were sanctioned by the Animal Experimentation Committee of the Kyoto Prefectural University of Medicine (M2022-86, M2021-56), adhering to the ARRIVE guidelines. Male B6.129P2-Apoetm1Unc/J (ApoEKO) mice were procured from Jackson Laboratory (Bar Harbor, ME, USA). ApoEST2DKO mice were generated by crossing ApoE KO mice with ST2 KO mice, sourced from The Institute of Medical Science of Osaka University [12] and bred in the University’s specific pathogen-free room. Mice were genotyped to determine that they were ApoEKO and ApoEST2DKO mice (refer to Appendix A). At the outset of the experimental regimen, ApoEKO and ApoEST2DKO mice were 16 weeks old. They had ad libitum access to a high-fat, high-sucrose diet (HFHSD; comprising 20% protein, 40% carbohydrate, 40% fat, coconut oil, and 0.3 g NaCl; D12327, Research Diets, Inc., New Brunswick, NJ, USA) for 8 weeks. The mice were housed in a controlled environment (temperature: 23 ± 1.5 °C; humidity: 40–60%; under a 12 h light/dark cycle from 7 a.m. to 7 p.m.). Cumulative oral intake was monitored over the course of the 8 weeks.

Fresh food, weighed manually, was provided to each cage every third day at 9:00 a.m. The amount of food consumed by each mouse per day was measured, and the leftover food was discarded. Upon reaching 24 weeks of age, the mice underwent overnight fasting for 16 h and were euthanized using a combination of anesthetic agents: 4.0 mg/kg midazolam, 0.3 mg/kg medetomidine, and 5.0 mg/kg butorphanol [15]. Euthanasia involved puncturing the left ventricle to collect blood in a heparinized syringe to prevent clotting. Subsequently, to eliminate circulating blood, the mice were gently perfused with 10 mL of isotonic sodium chloride solution via the left ventricle, after which adipose tissues surrounding the bilateral epididymis, jejunum, and aorta were dissected.

### 2.2. Blood Pressure Assessment

Systolic and diastolic blood pressure were evaluated non-invasively while keeping the mice warm using a mouse non-observational blood pressure measuring device (mice tail-cuff sphygmomanometer) (BP-98AL V3.02; Softron Corporation, Tokyo, Japan). Mean arterial pressure was computed by employing the following formula: mean = 1/3 of diastolic pressure + pulse pressure [16,17]. To validate the results, blood pressure was measured three times per mouse, and the average value was calculated.

### 2.3. Hematological Analysis

Hematological specimens were obtained from mice subjected to fasting, and serum was isolated through centrifugation at 15,000× *g* for 10 min at 4 °C. Triglycerides (TG), total cholesterol (T-Chol), low-density lipoprotein (LDL) cholesterol, high-density lipoprotein (HDL) cholesterol, and non-esterified fatty acids (NEFAs) were quantified by employing enzymatic techniques. Biochemical analyses were conducted utilizing FUJIFILM Wako Pure Chemical Corporation (Osaka, Japan) reagents.

### 2.4. Measurement of Arteriosclerotic Lesion Area

Atherosclerotic lesions within the aortic root were utilized for quantifying the arteriosclerotic area. Aortas, harvested by employing established methodologies detailed in the previous literature, were perfused with phosphate-buffered saline solution, then embedded in cryogenic tissue-embedding medium, snap-frozen in dry ice [18], and sectioned at 30 μm intervals until the aortic valve was discernible using a cryostat. Sections were then adjusted to a thickness of 10 μm, and consecutive segments (10 sections per sample) were scrutinized.

The tissues were fixed in 60% isopropanol for 15 s and stained with Oil Red O (Wako Pure Chemicals, Osaka, Japan) for 30 min at ambient temperature (23 ± 1.5 °C). Following staining, images were captured utilizing a BZ-X710 microscope (Keyence Co., Osaka, Japan), and the area of arteriosclerotic stiffness was quantified utilizing ImageJ software (version 1.53 k; National Institutes of Health, Bethesda, MD, USA).

### 2.5. Isolation of Mononuclear Cells from Aortas in Mice

Adipose tissue proximal to the adventitia was meticulously dissected and excised, ensuring preservation of the outer aortic membrane integrity. Adjacent lymph nodes were delicately excised.

The entire aorta was procured, and the atherosclerotic plaque was disengaged from the intima layer. Aortic segments were placed in 60 mm Petri dishes containing ice-cold fluorescence-activated cell-sorting (FACS) buffer and stored until enzymatic digestion. In a 50 mL Falcon tube, 5 mL of Roswell Park Memorial Institute (RPMI) buffer was added, and an aortic tissue piece was placed in the tube. The tissue was transferred to a dish, the RPMI buffer was aspirated and discarded, and the aortic tissue was cut into small pieces with surgical scissors.

In addition, 4 mL Hanks’ buffered salt solution and 1 mL Collagenase I were added. The tissues were cut into small pieces and were then transferred to a 37 °C incubator for 30 min with gentle agitation. Subsequently, the digested solution was strained through a 70 μm cell strainer positioned atop a fresh 50 mL Falcon tube. Any remaining aortic tissue was macerated using a syringe plunger, and the cell strainer was rinsed with 5 mL of FACS buffer. The filtrate was collected and subjected to centrifugation at 300× *g* for 4 min at 4 °C. The resultant supernatant was meticulously decanted, and the cellular pellet was resuspended in 400 μL of FACS buffer [19].

### 2.6. Tissue Preparation and Flow Cytometry

Stained cells underwent analysis utilizing FACS Canto II, and the resultant data were processed utilizing FlowJo version 10 software (Ashland, OR, USA). The following antibodies procured from eBioscience (Thermo Fisher Scientific, Waltham, MA, USA) were employed for the delineation of innate lymphoid cells: Biotin-CD3e (100304; clone: 145-2C11; 1/200), Biotin-CD45R/B220 (103204; clone: RA3–6B2; 1/200), Biotin-Gr-1 (108404; clone: RB6-8C5; 1/200), Biotin-CD11c (117304; clone: N418; 1/200), Biotin-CD11b (101204; clone: M1/70; 1/200), Biotin-Ter119 (116204; clone: TER-119; 1/200), Biotin-FceRIa (134304; clone: MAR-1; 1/200), FITC-Streptavidin (405202; 1/500), PE-Cy7-CD127 (135014; clone: A7R34; 1/100), Pacific Blue-CD45 (103116; clone: 30-F11; 1/100), PE-GATA-3 (clone: TWAJ; 1/50), APC-RORγ (clone: AFKJS-9; 1/50), and Fixable Viability Dye eFluor 780 (1/400) [20,21]. Additionally, the following antibodies (eBioscience, San Diego, CA, USA) were utilized for the discrimination of M1 and M2 macrophages: APC-CD45.2 (17045482; clone: 104; 1/50), PE-F4/80 (12480182; clone: BM8; 1/50), APC-Cy7-CD11b (47011282; clone: M1/70; 1/50), FITC-CD206 (MA516870; clone: MR5D3; 1/50), and PE-Cy7-CD11c (25011482; clone: N418; 1/50) [22].

### 2.7. Quantification of Free Fatty Acids in the Aorta and Sera

The fatty acid compositions present in the aorta and sera of ApoEKO and ApoEST2DKO mice were quantified utilizing gas chromatography–mass spectrometry (GC-MS), employing an Agilent 7890B/7000D instrument (Agilent Technologies, Santa Clara, CA, USA). A total of 10 mg of aortic tissue and 50 μL of serum were subjected to methylation using a fatty acid methylation kit (Nacalai Tesque Inc., Kyoto, Japan). The resultant product was then introduced onto a Varian capillary column (DB-FATWAX UI, Agilent Technologies, Santa Clara, CA, USA).

For fatty acid separation, the CP-Sil 88 FAME capillary column was employed (with dimensions of 100 m × an inner diameter of 0.25 mm × membrane thickness of 0.20 μm; Agilent Technologies). The temperature of the column was initially set at 100 °C for a duration of 4 min, followed by a gradual increase of 3 °C per minute until reaching 240 °C, where it was maintained for 7 min. Sample injection was performed in split mode with a split ratio of 5:1. Each fatty acid methyl ester was then detected in the selected ion-monitoring mode. All obtained results were subsequently standardized against the peak height of the C17:0 internal standard [23].

### 2.8. Quantitative Real-Time Polymerase Chain Reaction of the Aorta and Jejunum

Gene expression analysis was conducted by employing quantitative real-time polymerase chain reaction (q-RT-PCR).

Total RNA isolated from the aorta and jejunum underwent quantification using Thermo Scientific™ NanoDrop Lite (Thermo Fisher Scientific). Subsequently, all samples were diluted to a concentration of 5 ng/μL with DNase–RNase-free water. The relative expression levels of each target gene in the aorta (*Mcp1(Ccl2)*, *Il1b*, *Ifng*, *Tnfa*, *Il33*, *Il4*, *Il5*, *Icam1*, and *Vcam1*), jejunum (*Cd36*, *Il6*, *Pept1*, *Sglt1*, and *Tnfa*), and epididymal white adipose tissue (eWAT) (*Mcp1(Ccl2)*, *Il1b*, *Ifng*, and *Tnfa*) were normalized by utilizing the *Gapdh* threshold cycle (CT) values and quantified by employing the comparative threshold cycle 2^−ΔΔCT^ method, as previously outlined [24].

Given the predominant site of fatty acid absorption in the small intestine, we examined *Cd36* and *Il22* gene expression levels in the jejunum instead of the large intestine, aiming to explore alterations in fatty acid absorption. Nine mice from each group were assessed, with RT-PCR conducted in triplicate for each sample.

### 2.9. Cells and Treatment

Human aortic endothelial cell basal medium (cat. 22221) was purchased from Clontech TaKaRa cellartis (Tokyo, Japan). Human aortic endothelial cells (HAECs; cat. C0065C) and Low Serum Growth Supplement (LSGS; cat. S00310) were obtained from Thermo Fisher Scientific K.K (Tokyo, Japan).

LSGS comprises fetal bovine serum, basic fibroblast growth factor, heparin, hydrocortisone, and epidermal growth factor. LSGS was diluted 50-fold, administered in human aortic endothelial cell basal medium, and supplemented with 1% penicillin/streptomycin antibiotic solution; endothelial cells were cultured in a humidified atmosphere at 37 °C and 5% CO_2_. The culture medium was refreshed every 48 h. HAECs were utilized between passages 3–11.

### 2.10. qRT-PCR

HAECs (4 × 10^4^ cells/well) were plated in 96-well plates. Following adhesion, HAECs were either left untreated or treated with 0.1 mM palmitate (PA, a cytotoxic saturated fatty acid) [25] or with different doses of IL-33 (50 or 200 ng/mL) for 24 h. The supernatant in the plate was discarded, 1 mL of phosphate-buffered saline was added per well, and the cells were removed from the plate using a cell scraper. qRT-PCR was utilized to assess the mRNA expressions of *Gapdh*, *SREBF-1*, *FASN*, and *SCD-1* in cells.

Total RNA was extracted from the cells and reverse transcribed using the High-Capacity cDNA Reverse Transcription Kit (Applied Biosystems, Waltham, MA, USA), employing oligonucleotide dT primers and random hexamer primers, following the manufacturer’s recommendations. First-strand cDNA synthesis was conducted, with reverse transcription reactions carried out at 37 °C for 120 min, followed by enzyme inactivation through incubation at 85 °C for 5 min. RT-PCR was conducted using TaqMan Fast Advanced Master Mix (Applied Biosystems, Inc., Waltham, MA, USA), as per the manufacturer’s instructions. The PCR conditions employed were as follows: 1 cycle of 2 min at 50 °C and 20 s at 95 °C, followed by 40 cycles of 1 s at 95 °C and 20 s at 60 °C.

### 2.11. Histological Examination

Oil Red O staining was employed for the evaluation of intracellular lipid accumulation. To gauge HAEC adipogenesis, subsequent to the removal of the staining solution, the dye trapped within the cells was extracted using 200 μL of isopropanol and quantified utilizing the Image Pro Plus software (Ver.10) [26].

### 2.12. Statistical Analyses

The data underwent analysis utilizing JMP version 14.0 software (SAS, Cary, NC, USA). Disparities between the two groups were assessed by employing Student’s *t*-test. Statistical significance was predetermined at *p* < 0.05, with asterisks denoting statistical significance in the figures as follows: * *p* < 0.05, ** *p* < 0.01, *** *p* < 0.001, and **** *p* < 0.0001. Comparisons of three or more groups were analyzed using 1-way ANOVA multiple comparison tests. Statistical significance concerning body weight and oral intake was evaluated through two-way repeated measures analysis of variance followed by Bonferroni’s tests. The figures were created using the GraphPad Prism software (version 9.0; San Diego, CA, USA).

## 3. Results

### 3.1. Comparison of Body Weight and Serum Lipid Levels between ApoEKO and ApoEST2DKO Mice

An HFHSD was administered to ApoEKO and ApoEST2DKO mice, and various physiological parameters including body weight, blood pressure, and serum lipid profiles were assessed. Body weight (depicted in Figure 1A) and food intake (illustrated in Figure 1B) exhibited comparable trends between the two groups. However, systolic blood pressure demonstrated a significant elevation in ApoEST2DKO mice. Conversely, diastolic blood pressure, mean arterial pressure, and pulse pressure manifested similar patterns between the two cohorts (presented in Figure 1C–F).

An analysis of serum lipid profiles unveiled higher levels of serum triglycerides (TG), total cholesterol (T-chol), low-density lipoprotein cholesterol (LDL-chol), and non-esterified fatty acids (NEFAs) in the ApoEST2DKO group compared to the ApoEKO group. Conversely, serum high-density lipoprotein cholesterol (HDL-chol) levels were observed to be lower in the ApoEST2DKO group in contrast to the ApoEKO group (as shown in Figure 1G–K).

### 3.2. Comparison of Atherosclerotic Foci in the Aortic Rings of ApoEKO and ApoEST2DKO Mice Treated with an HFHSD

The extent of atherosclerotic lesions in the aortic sinus was quantified (depicted in Figure 1L). It was observed that the atherosclerotic area was notably greater in the ApoEST2DKO group compared to the ApoEKO group (as illustrated in Figure 1M).

### 3.3. Comparison of Visceral Fat Mass Weight in ApoEKO and ApoEST2DKO Mice Treated with an HFHSD

Fat around the epididymis was excised and weighed to assess visceral fat mass (refer to Figure 1N,O). Epididymal fat weight increased in the ApoEST2DKO group. The ratio of epididymal fat weight to total body weight did not show significant disparity between the two groups.

### 3.4. Aortic Inflammation and Anti-Inflammatory Cell Populations

We conducted a flow cytometric analysis in both groups: The ratio of M1/M2 macrophages in the aortas of the ApoEST2DKO group surpassed those in the ApoEKO group (Figure 2A). The proportion of ILC1 in CD45+ cells was elevated, whereas that of ILC2 was reduced (Figure 2C) in the aortas of the ApoEST2DKO group compared to those in the ApoEKO group (Figure 2B). Similar trends were observed in adipose tissue (Figure 2D–F).

### 3.5. Saturated Fatty Acids Present in the Aorta and Serum

An analysis conducted through GC-MS unveiled that concentrations of saturated fatty acids in both the serum and aorta of the ApoEST2DKO group surpassed those in the ApoEKO group. Specifically, myristic acid exhibited a notable elevation in both the serum and aorta of the ApoEST2DKO group (Figure 3A–G).

### 3.6. Expression of Genes Related to Inflammatory and Anti-Inflammatory Cytokines and Cell Adhesion Factors

The gene expressions of inflammatory and anti-inflammatory cytokines and cell adhesion factors were analyzed using qRT-PCR to determine the effect of HFHSD loading on chronic inflammation in the ApoEKO and ApoEST2DKO groups.

The aortas of the ApoEST2DKO group revealed significantly higher expression levels of genes such as *Mcp1 (Ccl2)*, *Il1b*, *Ifng*, and *Tnfa*, which are linked with inflammatory cytokines, compared to those in the ApoEKO group (Figure 4A–D).

Conversely, the gene expression levels of *Il33*, an activator of ILC2 (Figure 4E), and those associated with anti-inflammatory cytokines, such as *Il4* and *Il5*, were lower in the aortas of the ApoEST2DKO group than those in the ApoEKO group (Figure 4F,G). The levels of cell adhesion factors were markedly elevated in the ApoEST2DKO group compared to the ApoEKO group (Figure 4H,I).

### 3.7. eWAT Expression of Inflammatory Cytokine-Related Genes

The eWAT also showed increased levels of inflammatory cytokine-related genes in the ApoEST2DKO group compared to the ApoEKO group (Figure 4J–M).

### 3.8. Expression of Fatty Acid Transporters and Genes Related to Protein Metabolism, Sugar Metabolism, and Inflammatory Cytokines in the Jejunum

*Cd36* and *Il6* expressions in the jejunum tended to be non-significantly higher in the ApoEST2DKO group than those in the ApoEKO group (Figure 4N,O).

*Pept1* and *Sglt1* expressions were significantly increased (Figure 4P,Q), whereas *Tnfa* expression levels tended to increase in the ApoEST2DKO group (Figure 4R).

### 3.9. Evaluation of Fatty Acid Metabolism in Atherosclerosis Using Primary Normal HAECs

HAECs were treated with IL-33 and PA (200 μM/L), followed by analysis of the expressions of genes involved in fatty acid metabolism using q-RT-PCR.

To assess the differential expressions of fatty acid metabolism genes, HAECs were classified into four groups: no treatment (NC), only PA, PA plus IL-33 at a low concentration (50 ng/mL), and PA plus IL-33 at a high concentration (200 ng/mL).

The upregulation of genes involved in fatty acid metabolism was notably observed in the high IL-33 concentration group (Figure 5A–C). Thus, IL-33 treatment promotes the synthesis of saturated to unsaturated fatty acids, reducing saturated fatty acid levels.

Intracellular fat droplets were stained using an Oil Red O solution to determine the amount of cellular fat (Figure 5D–G). The area covered by Oil Red O staining was significantly reduced in the group receiving additional and particularly high IL-33 concentrations (Figure 5H). Thus, IL-33 decreases saturated fatty acid levels.

## 4. Discussion

Using ApoEKO and ApoEST2DKO mice, we demonstrated that on administering an HFHSD, IL33/ST2 signaling deficiency enlarged atherosclerotic foci, increased inflammatory gene expression levels, and significantly increased saturated fatty acid concentrations within the atherosclerotic foci. ApoE (apolipoprotein E) transports lipids in the body and plays a vital role in plasma lipid equilibrium. High-fat diet administration in ApoEKO mice predisposes them to metabolic syndrome and non-alcoholic steatohepatitis [27,28]. ST2, a component of the IL-1 family, exists in both transmembrane (ST2L) and soluble (sST2) forms. ST2L is an IL-33 receptor, and IL-33/ST2 signaling produces inflammatory cytokines and chemokines, which induce immune responses [29]. In our study, we observe an increased M1/M2 macrophage ratio and increased ILC1 and decreased ILC2 in both arteries and eWAT in mice lacking ApoEST2. In ApoEKO mice, the expressions of cell adhesion molecules, vascular cell adhesion molecule 1 (VCAM-1), and intercellular adhesion molecule 1 (ICAM-1) are increased, and the enhanced VCAM-1 expression, in particular, is responsible for macrophage accumulation in atherosclerotic nests [30]. Moreover, IL-4 is essential for inhibiting atherosclerosis progression [30]. Macrophages activated via the Wnt pathway secrete prostaglandin E2, which leads to the activation of signal transducers and the activator of transcription (STAT) 3. It then associates with IL-4/STAT6 to stimulate plaque macrophages and inhibit atherosclerosis progression [31]. Atherosclerosis development has also been reported in mice lacking IL-5 production [32]. In this study, mice in the ApoEST2DKO group, with significant atherosclerosis development, showed increased VCAM1 and ICAM1 levels and a non-significant but decreasing trend in IL-4 and IL-5, consistent with previously reported results. Although not reaching hypertensive levels, the reason for the increased systolic blood pressure in the ApoEST2DKO mice group compared to the ApoEKO mice group could be worsening arterial stiffness. Mice subjected to a high-fat regimen alongside Bacteroides fragilis exhibited heightened levels of total cholesterol and LDL-c in their circulatory system, alongside a notable augmentation in atherosclerotic lesions. Concurrently, there was an observable decline in *Lactobacillus* and a rise in *Desulfovibrionaceae* within the intestinal microbiota. Furthermore, there was a marked increase in the mRNA expression of inflammatory cytokines *CD36* and F4/80 within the arterial walls and duodenum of these mice [33]. Other previous reports have shown diminished excretion of short-chain fatty acids and alterations in intestinal microbiota composition, including reduced levels of *Lachnospiraceae_FCS020*, *Ruminococcaceae_UCG-009*, *Acetatifactor*, *Lachnoclostridium*, and *Lactobacillus_gasseri*, in elderly mice afflicted with advanced atherosclerosis [34,35]. In our study, ApoEST2KO mice had altered expression of inflammatory genes (*CD36*, *Il6*, and *Tnfa*).

When stimulated by ox-LDL, vascular smooth muscle cells initiate the generation of reactive oxygen species through the *Cd36* signaling cascade and produce inflammatory cytokines. This pathway promotes intimal hyperplasia through vascular injury, involving a specific Src family kinase Fyn, which facilitates the phosphorylation and subsequent degradation of Nrf2, a transcription factor governing the expression of antioxidant genes [36]. CD36 also forms complexes with Toll-like receptor 4 (TLR4) and urokinase receptors in response to ox-LDL and produces inflammatory cytokines through NF-κB activation [37,38].

In the ApoEST2DKO mice in this study, arterial saturated fatty acid concentrations (myristic acid, palmitic acid, and stearic acid) were elevated. Saturated fatty acids exacerbate adipocyte–macrophage interactions (via TLR4 signaling, saturated fatty acids/TLR4/NF-KB signals) and inflammatory changes in the adipose tissue, disrupting the adipocytokine production regulation mechanism. The remodeling of adipose tissue induced by saturated fatty acids plays a pivotal role in the initiation and progression of atherosclerosis [39].

Hooper et al. have reported a 21% reduction in the incidence of cardiovascular events in humans subjects following a diminished consumption of saturated fatty acids [40]. Increased LDL-C levels due to saturated fatty acid intake are associated with increased apo B levels; the presence of total, medium, and small LDL particles may be associated with atherogenic dyslipidemia [41]. Similarly, in mice, a high-fat diet increased atherosclerosis, leukocyte counts, inflammatory cytokine levels, and neutrophil accumulation in atherosclerotic nests while decreasing the concentration of anti-inflammatory short-chain fatty acids [42]. Thus, humans and mice may be at risk of atherosclerosis if they have increased saturated fatty acid levels in the body.

Relative to wild-type (WT) murine counterparts, ST2KO mice show increased inflammatory cytokine levels in white adipose tissue, apoptosis of macrophages, and increased levels of saturated fatty acids such as lauric, myristic, palmitic, and stearic acids within adipose tissue [12]. The external administration of IL-33 to WT and ST2KO mice improves fatty acid metabolism in the adipose tissue [12]. In addition, IL-33 secreted from ILC2 induces Th2 cytokines and ox-LDL antibodies and inhibits atherosclerosis development [11,43]. The use of HAECs to elucidate the relationship between atherosclerosis and IL-33 revealed a significant increase in arterial IL-33 levels in ApoEKO mice.

HAECs treated with PA and IL-33 demonstrated significantly higher expressions of fatty acid metabolism genes compared to HAECs treated with PA alone. Stearoyl-CoA desaturase catalyzes the synthesis of monounsaturated fatty acids from saturated fatty acid precursors and serves as a pivotal enzyme in the fatty acid composition of cellular lipids [44]. *Srebf1* (sterol regulatory sequence binding protein 1) and *Fasn* (fatty acid synthase) genes are involved in regulating fatty acid synthesis [45]. SREBP1 is a transcription factor inducing the *Srebf1* gene; polyunsaturated fatty acids attenuate SREBP1 mRNA expression [46]. The upregulation of these fatty acid metabolism genes in this study was due to IL-33 administration, suggesting that IL-33 may promote the metabolism from saturated to unsaturated fatty acids. Moreover, the area was measured using Oil Red O staining, which specifically stains lipids. The staining area was lesser in HAECs treated with PA and IL-33 compared to the group treated with PA alone, particularly in those with high IL-33 concentrations. Thus, a potential antiatherosclerosis mechanism of ILC2 is to prevent saturated fatty acid accumulation by IL-33.

There are limitations to this study. It is possible that ST2 signaling in endothelial cells has some effect on NO signaling, but we have not been able to measure NO in our study. Further pathways by which IL-33 promotes the metabolism of saturated fatty acids to unsaturated fatty acids have not been elucidated. In vivo supplementation of IL-33 to reverse atherosclerosis is a topic for future research to clarify the validity of this study.

## 5. Conclusions

An atherosclerosis mouse model with further attenuated ILC2 showed worsened atherosclerosis, increased gene expression levels of inflammatory cytokines in the atherosclerotic foci, and increased saturated fatty acid levels in the aorta and serum. Exogenous administration of saturated fatty acids and IL-33 to HAECs in vitro significantly decreased the fatty acid area in the aortic cells. This study demonstrates an atherosclerosis-preventive mechanism of ILC2, attenuating saturated fatty acid accumulation by IL-33.

## Figures and Tables

**Figure 1 nutrients-16-01195-f001:**
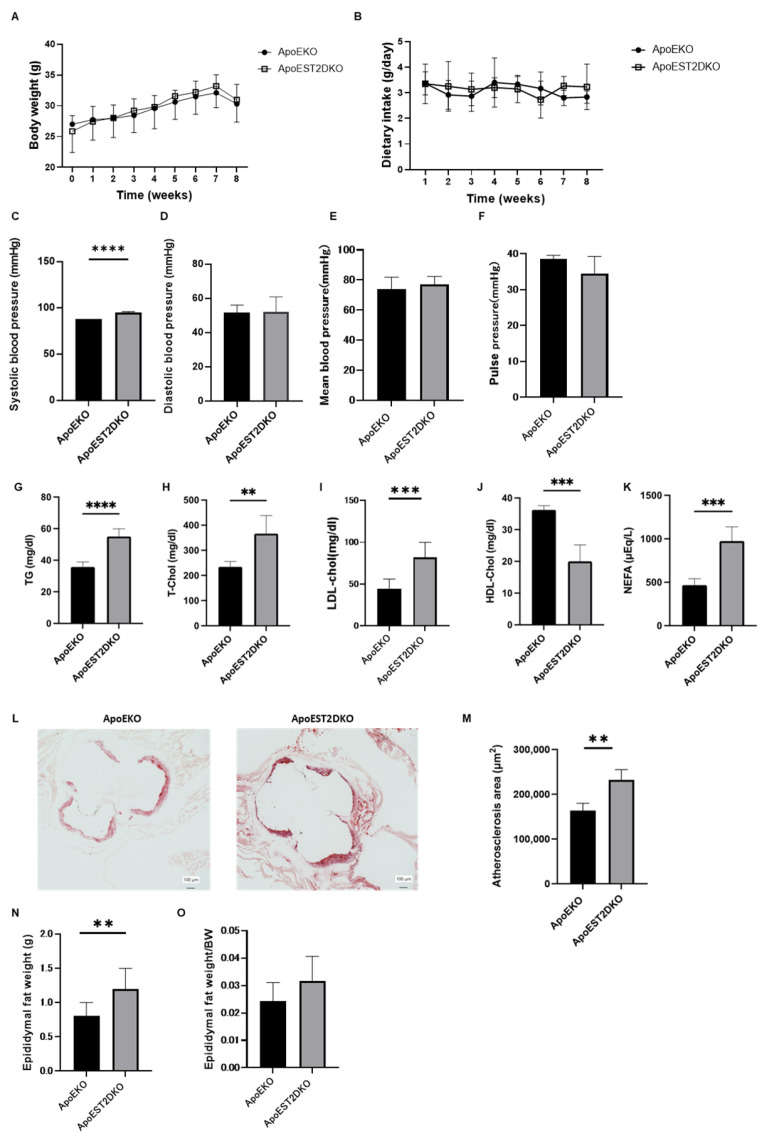
Alterations in body weight, dietary intake, hematological parameters, arterial stiffness, and visceral adiposity in a murine model of atherosclerosis (ApoEKO) and an atherosclerosis model deficient in the IL-33 receptor, ST2 (ApoEST2DKO. (**A**) Changes in body weight in ApoE KO and ApoEST2 DKO mice at 16 and 24 weeks fed an HFHSD (n = 9). Analysis conducted through two-way repeated measures analysis of variance (ANOVA) followed by Bonferroni’s tests (n = 9). (**B**) Dietary intake (n = 9). (**C**) Systolic blood pressure (n = 9). (**D**) Diastolic blood pressure (n = 9). (**E**) Mean arterial pressure (n = 9). (**F**) Pulse pressure (n = 9). Serum concentrations of (**G**) TG (n = 9), (**H**) T-Chol (n = 9), (**I**) LDL-Chol (n = 9), (**J**) HDL-Chol (n = 9), and (**K**) NEFAs (n = 9). (**L**) Representative histological images of aortic valves stained with Oil Red O (n = 9). (**M**) Atherosclerotic lesion area (**N**,**O**). Absolute and relative epididymal fat mass. Data are expressed as mean ± standard deviation (SD) and analyzed using unpaired *t*-tests: ** *p* < 0.01, *** *p* < 0.001, and **** *p* < 0.0001. HDL-Chol, high-density lipoprotein cholesterol; HFHSD, high-fat, high-sucrose diet; LDL-Chol, low-density lipoprotein cholesterol; NEFA, non-esterified fatty acid; TG, triglycerides; T-Chol, total cholesterol.

**Figure 2 nutrients-16-01195-f002:**
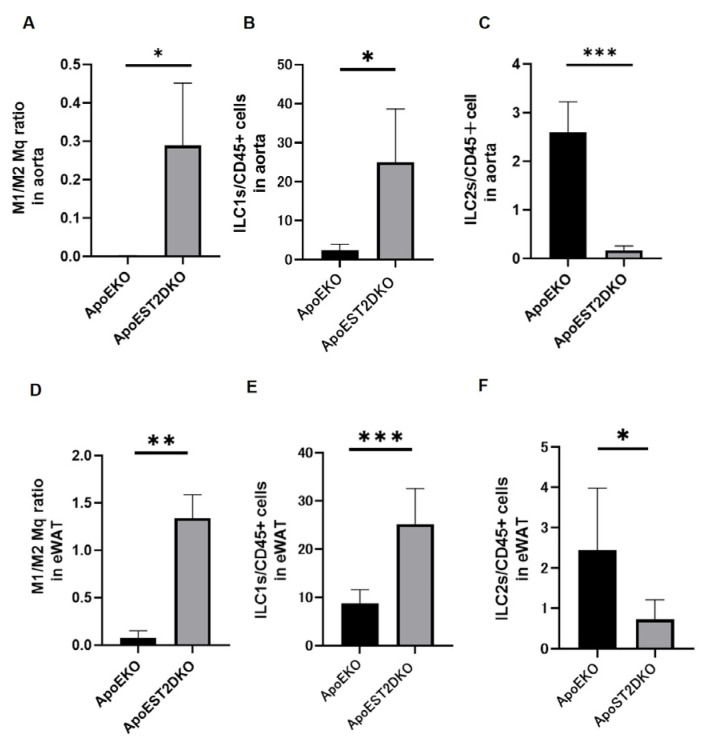
The composition of innate immune cell populations within the aorta and epididymal white adipose tissue (eWAT) in both ApoEKO and ApoEST2DKO mice. (**A**) The ratio of M1/M2 macrophages (n = 9), (**B**) the ratio of ILC1s/CD45-positive cells (n = 9), and (**C**) the proportion of ILC2s/CD45-positive cells within the aorta are depicted (n = 9). (**D**) Likewise, the ratio of M1/M2 macrophages (n = 9), (**E**) the proportion of ILC1s/CD45-positive cells (n = 9), and (**F**) the proportion of ILC2s/CD45-positive cells within the eWAT are presented (n = 9). The data are expressed as mean values with standard deviation and were analyzed utilizing unpaired *t*-tests, with significance levels denoted as * *p* < 0.05, ** *p* < 0.01, and *** *p* < 0.001.

**Figure 3 nutrients-16-01195-f003:**
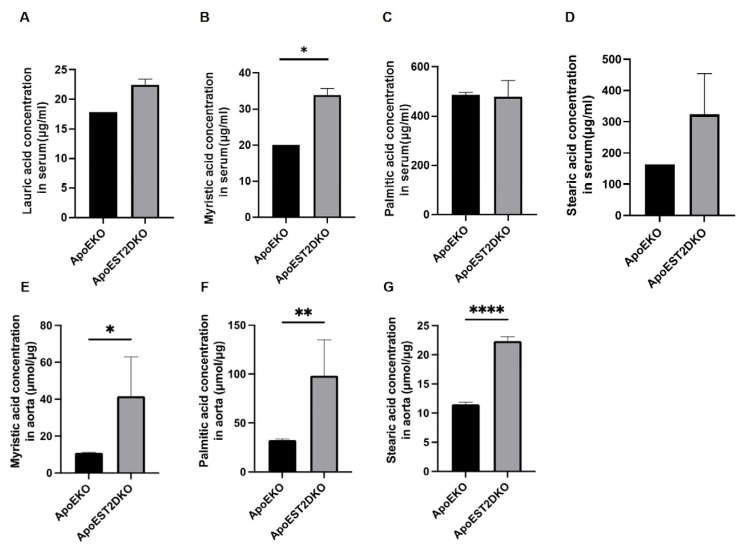
The concentrations of saturated fatty acids in the sera and aorta of ApoEKO and ApoEST2DKO mice. The graphs in (**A**–**D**) depict the concentrations of lauric, myristic, palmitic, and stearic acids in the sera (n = 9), while (**E**–**G**) showcases the concentrations of myristic, palmitic, and stearic acids in the aorta (n = 9). The data are presented as mean values with standard deviation and were subjected to analysis using unpaired *t*-tests; significance levels are denoted as * *p* < 0.05, ** *p* < 0.01, and **** *p* < 0.0001.

**Figure 4 nutrients-16-01195-f004:**
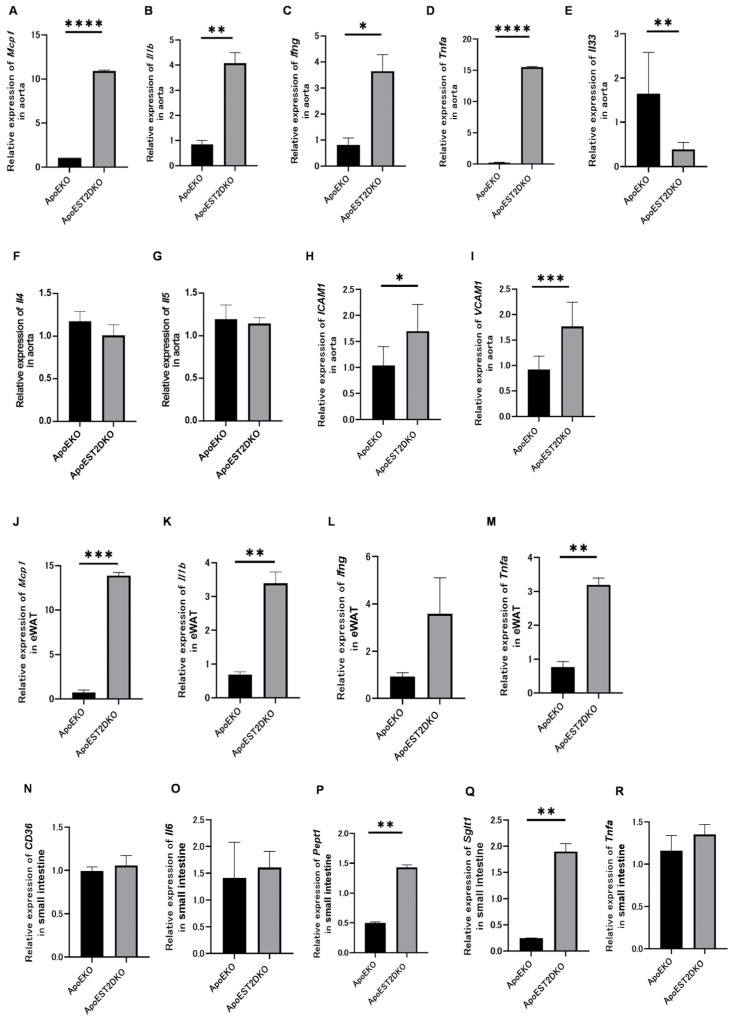
Relative mRNA expression levels: (**A**) Mcp1 (Ccl2) (n = 9), (**B**) Il1b (n = 9), (**C**) Ifng (n = 9), (**D**) Tnfa (n = 9), (**E**) Il33 (n = 9), (**F**) Il4 (n = 9), (**G**) Il5 (n = 9), (**H**) ICAM1 (n = 9), and (**I**) VCAM1 in the aorta (n = 9); (**J**) MCP1 (Ccl2) (n = 9), (**K**) Il1b (n = 9), (**L**) Ifng (n = 9), and (**M**) Tnfa in the Ewat (n = 9); (**N**) CD36 (n = 9), (**O**) Il6 (n = 9), (**P**) Pept1 (n = 9), (**Q**) Sglt1 (n = 9), and (**R**) Tnfa in the jejunum, normalized to Gapdh expression (n = 9). The data are presented as mean ± standard deviation values and subjected to analysis using unpaired *t*-tests; significance levels are denoted as * *p* < 0.05, ** *p* < 0.01, *** *p* < 0.001, and **** *p* < 0.0001.

**Figure 5 nutrients-16-01195-f005:**
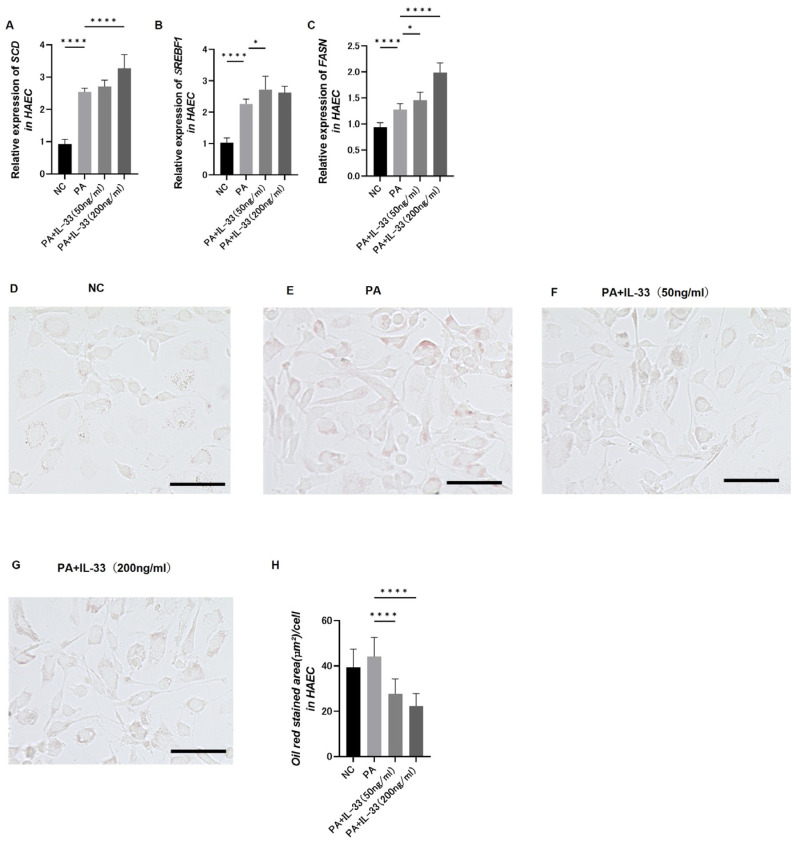
Expression levels of genes pertinent to fatty acid metabolism in human aortic endothelial cells (HAECs) classified into the following four groups: normal control, palmitic acid (PA, 200 μM/L), IL-33 (50 ng/mL) on PA, and IL-33 (200 ng/mL) on PA. Relative mRNA expressions of (**A**) Scd (n = 6), (**B**) Srebf1 (n = 6), and (**C**) Fasn (n = 6) in HAECs. (**D**–**G**) Oil Red O staining area per cells (n = 14). Stained with Oil Red O in HAECs: (**D**) NC group in HAECs (×20), scale bar: 100 μm; (**E**) Group with PA(200 μM/L) added in HAECs (×20), scale bar: 100 μm; (**F**) Group with PA (200 μM/L) and IL-33 (50 ng/mL) added in HAECs (×20), scale bar: 100 μm; (**G**) Group with PA(200 μM/L) and IL-33 (200 ng/mL) added in HAECs (×20), scale bar: 100 μm. (**H**) The data are presented as mean ± standard deviation values and subjected to analysis using 1-way ANOVA multiple comparison tests; significance levels are denoted as * *p* < 0.05, and **** *p* < 0.0001.

## Data Availability

The original contributions presented in the study are included in the article/Appendix A, further inquiries can be directed to the corresponding author.

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
