# Peer review of "IL-33 Reduces Saturated Fatty Acid Accumulation in Mouse Atherosclerotic Foci"

_nutrients, 2024, doi:10.3390/nu16081195_

Round 1

Reviewer 1 Report

Comments and Suggestions for Authors

This paper explores the mechanism of ILC2 to prevent atherosclerosis through IL-33, which is a meaningful study, as well as a well-written article. But I have some questions as follows.

1.     In line 44, since ILC 1, 2 and 3 were mentioned, and functions of ILC 1 and 2 were further introduced. Curiously, the main function of ILC3 may be better to be briefly added.

2.     Why choose mice lacking the IL-33 receptor in the first place? Is there a previous research basis for the effect of IL-33 on atherosclerosis? IL-33 may play a vital role in atherosclerotic disease, but it is hard to say this study regarding IL-33 can answer the question in line 69: make it clear that the cellular and molecular mechanism of atherosclerotic disease.

3.     Please introduce more about the relationship between saturated fatty and immunity? Personally think that a firm relationship between intestinal flora and fatty acid content. Thus gut microbiota measurement may be a good way to illustrate more about the role of IL-33 in atherosclerotic.

Author Response

Prof. Dr. Maria Luz Fernandez

Prof. Dr. Liuis Serra-Majem

Editor-in-Chief

Nutrients MDPI

                           14 Apr 2024

Dear Editor,

Ref.: Manuscript ID: nutrients-2930866

 Thank you for your kind letter concerning our manuscript.

Enclosed please find our revised manuscript entitled “IL-33 reduces saturated fatty acid accumulation in mouse atherosclerotic foci”, manuscript ID of which is nutrients-2930866.

 At first, we would like to thank reviewers for their constructive comments on our manuscript. Based on the reviewers’ comments, we have revised our manuscript. The responses to reviewers’ comments are described below. Your kind consideration of this paper would be greatly appreciated.

Response to Reviewer 1

Major comments

Comment 1

 *   In line 44, since ILC 1, 2 and 3 were mentioned, and functions of ILC 1 and 2 were further introduced. Curiously, the main function of ILC3 may be better to be briefly added.

Answer 1

 Thank you very much for your comment. As you pointed out, I have added a note to the introduction section about the functionality of ILC3.

Introduction(Page 2)

ILC1 secretes interferon (IFN)-γ and exhibits antimicrobial activity against intracel-lular microorganisms; ILC2 produces interleukin (IL)-5, IL-9, and IL-13, contributing to defense against parasite defense and allergic diseases; ILC3 generates IL-22 in response to IL-23 and IL-1β, participating in innate immunity against fungi and extracellular pathogens [a].

References

[a] Clottu, A.S.; Humbel, M.; Fluder, N.; Karampetsou, M.P.; Comte, D. Innate lymphoid cells in autoimmune diseases. Front. Immunol. 2021, 12, 789788. https://doi.org/10.3389/fimmu.2021.789788.

Comment 2

 * Why choose mice lacking the IL-33 receptor in the first place? Is there a previous research basis for the effect of IL-33 on atherosclerosis? IL-33 may play a vital role in atherosclerotic disease, but it is hard to say this study regarding IL-33 can answer the question in line 69: make it clear that the cellular and molecular mechanism of atherosclerotic disease.

Answer 2

 Thank you for your valuable comment. In a previous report, IL-33 subcutaneously injected twice weekly into ApoEKO mice markedly increased levels of IL-4, -5, and -13, decreased levels of IFNγ in serum and lymph node cells, and decreased atherosclerotic lesion area in the aortic sinus, compared to control mice [a]. From previous reports, IL-33 may show a protective role in atherosclerotic lesions [b], and indeed mice lacking IL-33 showed more exacerbation of atherosclerotic plaque in the aortic sinus than atherosclerotic model mice (ApoEKO mice) (Figure 1-M) by using mice lacking IL-33, I believe that IL-33 more robustly demonstrated that IL-33 has an inhibitory effect on atherosclerotic changes.

The introduction is being revised.

References

[a] Miller, A.M.; Xu, D.; Asquith, D.L.; Denby, L.; Li, Y.; Sattar, N.; Baker, A.H.; McInnes, I.B.; Liew, F.Y. IL-33 Reduces the Development of Atherosclerosis. Journal of Experimental Medicine 2008, 205, 339–346, doi:10.1084/JEM.20071868.

[b] Buckley, M.L.; Williams, J.O.; Chan, Y.-H.; Laubertová, L.; Gallagher, H.; Moss, J.W.E.; Ramji, D.P. The Interleukin-33-Mediated Inhibition of Expression of Two Key Genes Implicated in Atherosclerosis in Human Macrophages Requires MAP Kinase, Phosphoinositide 3-Kinase and Nuclear Factor-ΚB Signaling Pathways., doi:10.1038/s41598-019-47620-8.

Introduction(Page 2)

The exogenous administration of IL-33 to ApoEKO mice induced Th1-Th2 production in vivo and reduced ox-LDL antibody levels. This report also reported that exogenous administration of IL-33 to ApoEKO mice markedly increased levels of IL-4, -5, and -13, decreased levels of IFNγ in serum and lymph node cells, and decreased atherosclerotic lesion area in the aortic sinus [a]. Moreover, IL-33 promotes the browning of adipose tissue, prevents obesity, and decreases ILC2 levels in the adipose tissue of obese mice [b]. IL-33 may show a protective role in atherosclerotic lesions[c].

References

[a] Miller, A.M.; Xu, D.; Asquith, D.L.; Denby, L.; Li, Y.; Sattar, N. et al. IL-33 reduces the development of atherosclerosis. J. Exp. Med 2008, 205, 339–46. https://doi.org/10.1084/jem.20071868.

[b] Okamura, T.; Hashimoto, Y.; Mori, J.; Yamaguchi, M.; Majima, S.; Senmaru, T. et al. ILC2s improve glucose metabolism through the control of saturated fatty acid absorption within visceral fat. Front. Immunol. 2021, 12, 669629. https://doi.org/10.3389/fimmu.2021.669629. eCollection 2021.

[c] Buckley, M.L.; Williams, J.O.; Chan, Y.-H.; Laubertová, L.; Gallagher, H.; Moss, J.W.E.; Ramji, D.P. The Interleu-kin-33-Mediated Inhibition of Expression of Two Key Genes Implicated in Atherosclerosis in Human Macrophages Requires MAP Kinase, Phosphoinositide 3-Kinase and Nuclear Factor-ΚB Signaling Pathways., https//doi:10.1038/s41598-019-47620-8.

Comment 3

 * Please introduce more about the relationship between saturated fatty and immunity? Personally think that a firm relationship between intestinal flora and fatty acid content. Thus gut microbiota measurement may be a good way to illustrate more about the role of IL-33 in atherosclerotic.

Answer 3

    Thank you for your comment. According to your comment, we have added the sentences described as below.

Discussion (Page 13)

Mice subjected to a high-fat regimen alongside Bacteroides fragilis exhibited heightened levels of total cholesterol and LDL-c in their circulatory system, alongside a notable augmentation in atherosclerotic lesions. Concurrently, there was an observable decline in Lactobacillus and a rise in Desulfovibrionaceae within the intestinal microbiota. Fur-thermore, there was a marked increase in the mRNA expression of inflammatory cyto-kines Cd36 and F4/80 within the arterial walls and duodenum of these mice [a]. Other previous reports have shown diminished excretion of short-chain fatty acids and alter-ations in intestinal microbiota composition, including reduced levels of Lachnospirace-ae_FCS020, Ruminococcaceae_UCG-009, Acetatifactor, Lachnoclostridium, and Lactobacil-lus_gasseri, in elderly mice afflicted with advanced atherosclerosis [b,c].

References

[a] Shi, G.; Lin, Y.; Wu, Y.; Zhou, J.; Cao, L.; Chen, J. et al. Bacteroides fragilis supplementation deteriorated metabolic dys-function, inflammation, and aorta atherosclerosis by inducing gut microbiota dysbiosis in animal model. Nutrients 2022, 14, 2199. https://doi.org/10.3390/nu14112199.

[b] Sun, Y.; Wu, D.; Zeng, W.; Chen, Y.; Guo, M.; Lu, B. et al. The role of intestinal dysbacteriosis-induced arachidonic acid metabolism disorder in inflammaging in atherosclerosis. Front. Cell. Infect. Microbiol. 2021, 11, 618265. https://doi.org/10.3389/fcimb.2021.618265.

[c] Magne, F.; Gotteland, M.; Gauthier, L.; Zazueta, A.; Pesoa, S.; Navarrete, P. et al. The Firmicutes/Bacteroidetes Ratio: A relevant marker of gut dysbiosis in obese patients? Nutrients 2020, 12, 1474. https://doi.org/10.3390/nu12051474.

Thank you for giving us the opportunity to strengthen our manuscript with your valuable comments. We have worked hard to incorporate your feedback and hope that these revisions persuade you to accept our submission.

 We hope the revised version is now suitable for publication and look forward to hearing from you.

Yours faithfully,

Masahide Hamaguchi, MD, PhD

Department of Endocrinology and Metabolism, Kyoto Prefectural University of Medicine, Graduate School of Medical Science

Address: 465 Kajii-cho, Kawaramachi-Hirokoji, Kamigyo-ku, Kyoto 602-8566, Japan

Fax: +81-75-252-3721, Tel: +81-75-251-5506

E-mail: mhama@koto.kpu-m.ac.jp

Reviewer 2 Report

Comments and Suggestions for Authors

Introduction. I suggest that the first part of the final paragraph, where the nomenclature of IL-33 was introduced be moved to the 3rd paragraph to have a better flow of IL33 focused ideas. There are also several sentences about macrophages that diverts the attention from the need to study IL33 and ILC2.

I am surprised that the BP measurements were lower than most reported values in ApoE KO mice. For example, Yang reported MAP to be around 130 mmHg (PMID 10559023), Haggerty reported a systolic pressure to be between 120-130 mmHg in mice untreated with AngII (PMID 26086817). 

How were the relative expression of genes calculated? If it's using ∆∆CT, there must be some mistake with the calculations as the value for the control, which I assume would be the ApoE KO mice, greatly deviates from 1 (See Fig 4B, C, F, G)

Since the animal model used was a global KO of ST2, non-immune responses are observed like changes in the expression of intestinal transporter for glucose and peptides. Can the authors provide insights on why these nutrient transports are increased?

Can the authors provide a validation for the ST2 KO? 

ST2 is highly expressed in endothelial cells and endothelial control of blood pressure is due to eNOS activity. Can the authors explore whether ST2 signaling in endothelial cells has any effect on NO signaling?

What was the reason why the ApoE mice was chosen compared to other mouse models of atherosclerosis? It has been shown that the LDLr KO mouse closely resembles the human condition compared to ApoE KO.

There are disconnections between the gene expression of lipogenic genes and the lipid content of HAEC treated with IL33. These genes promote synthesis of different fatty acids, not fatty acid degradation. Please correct Line 348-349.  

The authors are interchangeably using the ST2 KO mouse as ILC2 deficiency. However, ST2 is also present in other cell types. Thus, it should be IL33/ST2 signaling deficiency and not ILC2 deficiency.

In Line 387-388, the authors mention that the increased blood pressure (despite not reaching hypertensive levels) was due to arterial stiffness. I believe this is highly speculative because there was no data on pulse wave velocity. 

Line 3923-395 is also highly speculative and not supported by data. In Fig 4N, O and R, ST2 KO did change the expression of inflammatory genes.

Overall, the discussion needs more integration of the results. The way it is written at its current form reads more of a literature review. 

Can the authors provide the sample size in the figure legends?

Author Response

Prof. Dr. Maria Luz Fernandez

Prof. Dr. Liuis Serra-Majem

Editor-in-Chief

Nutrients MDPI

                          14 Apr 2024

Dear Editor,

Ref.: Manuscript ID: nutrients-2930866

 Thank you for your kind letter concerning our manuscript.

Enclosed please find our revised manuscript entitled “IL-33 reduces saturated fatty acid accumulation in mouse atherosclerotic foci”, manuscript ID of which is nutrients-2930866.

 At first, we would like to thank reviewers for their constructive comments on our manuscript. Based on the reviewers’ comments, we have revised our manuscript. The responses to reviewers’ comments are described below. Your kind consideration of this paper would be greatly appreciated.

Response to Reviewer 2

Major comments

Comment 1

 *  Introduction. I suggest that the first part of the final paragraph, where the nomenclature of IL-33 was introduced be moved to the 3rd paragraph to have a better flow of IL33 focused ideas. There are also several sentences about macrophages that diverts the attention from the need to study IL33 and ILC2.

Answer 1

 Thank you very much for your comment. As indicated, "IL-33 is a member of the IL-1 family, which includes IL-1β and IL-18, and has potent immunomodulatory functions" relocated to the beginning of the third paragraph. And I reduced the description of macrophages.

Introduction【Page 2】

IL-33 is a member of the IL-1 family, which includes IL-1β and IL-18, and has potent immunomodulatory functions [a]. IL-33 is localized in the nucleus of vascular endo-thelial and epithelial cells. Upon injury, IL-33 is rapidly translocated extracellularly, stimulating basophils, mast cells, and ILC2 to produce Th2 cytokines (involved in in-nate-type allergy), or stimulates Th2 with antigen, enhancing IL-5 and IL-13 production (involved in acquired-type allergy).

T cells infiltrate the vascular wall and respond to the presence of oxidized low-density lipoprotein (ox-LDL) with smooth muscle and endothelial cells, producing cytokines and inflammatory mediators, which, depending on their pheno-type, can promote plaque formation (Th1) or inhibit inflammatory changes (Th2) [b]

References

[a] Schmitz, J.; Owyang, A.; Oldham, E.; Song, Y.; Murphy, E.; McClanahan, T.K. et al. IL-33, an interleukin-1-like cytokine that signals via the IL-1 receptor-related protein ST2 and induces T helper type 2-associated cytokines. Immunity 2005, 23, 479–90. https://doi.org/10.1016/j.immuni.2005.09.015.

[b] Fernández-Gallego, N.; Castillo-González, R.; Méndez-Barbero, N.; López-Sanz, C.; Obeso, D.; Villaseñor, A. et al. The im-pact of type 2 immunity and allergic diseases in atherosclerosis. Allergy 2022, 77, 3249–66. https://doi.org/10.1111/all.15426.

Comment 2

 * I am surprised that the BP measurements were lower than most reported values in ApoE KO mice. For example, Yang reported MAP to be around 130 mmHg (PMID 10559023), Haggerty reported a systolic pressure to be between 120-130 mmHg in mice untreated with AngII (PMID 26086817).

Answer 2

 Thank you for your comment. Upon reviewing the previous reports, as indicated, the systolic blood pressure of ApoEKO mice is typically around 120-130mmHg. However, in our  study, each sample was measured three times, resulting in an average systolic blood pressure of 88mmHg for ApoEKO mice and 95mmHg for ApoEST2DKO mice. Matoba's group, using the same method as the authors, had a mean blood pressure of about 80 mmHg as did the authors. “Paracrine osteogenic signals via bone morphogenetic protein-2 accelerate the atherosclerotic intimal calcification in vivo”. Unlike other reports, the authors and Matoba's group measured blood pressure under anesthesia, and blood pressure is thought to be decreased by anesthesia.

Comment 3

 * How were the relative expression of genes calculated? If it's using ∆∆CT, there must be some mistake with the calculations as the value for the control, which I assume would be the ApoE KO mice, greatly deviates from 1 (See Fig 4B, C, F, G).

Answer 3

    Thank you for your comment. The calculations were performed using ∆∆∆CT, but some errors were made, so we would like to correct them. Replaced the figure of the part pointed out in the revised paper.

Comment 4

* Since the animal model used was a global KO of ST2, non-immune responses are observed like changes in the expression of intestinal transporter for glucose and peptides. Can the authors provide insights on why these nutrient transports are increased?

Answer 4

   Thank you for pointing that out. Pept1 and Sglt1 play crucial roles in nutrient absorption in the intestinal tract, and their increased activity is associated with excessive nutrient uptake. Excessive nutrient absorption leads to conditions such as obesity and metabolic disorders, which increase the risk of developing atherosclerosis. Furthermore, Pept1 and Sglt1 are involved in the uptake of nutrients such as glucose and amino acids, and their activity is heightened in conditions of high blood sugar and high fat levels. The upregulation of intestinal transporters for glucose and peptides may exacerbate inflammation and oxidative stress, potentially impairing endothelial function and causing degenerative changes in arterial walls. Therefore, the elevation of Pept1 and Sglt1 is likely to contribute to the progression of atherosclerosis through interconnected factors such as excessive nutrient intake, metabolic abnormalities, and associated inflammatory responses.

However, as it has not yet been fully elucidated, further investigation is deemed necessary regarding the mechanisms underlying the increased nutrient transport.

Comment 5

* Can the authors provide a validation for the ST2 KO?

Answer 5

Thank you for comment. We will send you PCR results of mouse genotyping. Please refer to the supplementary file.

Comment 6

   * ST2 is highly expressed in endothelial cells and endothelial control of blood pressure is due to eNOS activity. Can the authors explore whether ST2 signaling in endothelial cells has any effect on NO signaling?

Answer 6

    Thank you for pointing this out. We have not been able to measure NO in this study, but it is a very effective tool and will be the subject of future research. I added the following as a limitation.

Discussion(page14)

It is possible that ST2 signaling in endothelial cells has some effect on NO signaling, but we have not been able to measure NO in our study.

Comment 7

   * What was the reason why the ApoE mice was chosen compared to other mouse models of atherosclerosis? It has been shown that the LDLr KO mouse closely resembles the human condition compared to ApoE KO.

Answer 7

    Thank you for your valuable comment. As noted, while LDLr KO mice also serve as models for atherosclerosis, we believe that ApoE KO mice are more suitable for observing inflammatory cytokines and the progression of atherosclerosis. Firstly, ApoE is involved in regulating inflammatory responses, thus influencing the progression of atherosclerosis. Secondly, ApoE KO mice lack the ApoE gene, rendering them particularly sensitive to inflammatory responses and the progression of atherosclerosis induced by obesity or metabolic abnormalities. In contrast, LDLr KO mice exhibit hypercholesterolemia due to the absence of LDL receptors, which may not directly impact inflammatory responses or the progression of atherosclerosis as significantly as in ApoE KO mice.

Comment 8

 * There are disconnections between the gene expression of lipogenic genes and the lipid content of HAEC treated with IL33. These genes promote synthesis of different fatty acids, not fatty acid degradation. Please correct Line 348-349. 

Answer 8

 Thank you for your comment. As pointed out, I have corrected it as follows.

Results【3.9. Evaluation of fatty acid metabolism in atherosclerosis using primary normal HAECs [Page 11]】

Thus, IL-33 treatment promotes synthesis of saturated to unsaturated fatty acids, reducing saturated fatty acid levels.

Comment 9

    * The authors are interchangeably using the ST2 KO mouse as ILC2 deficiency. However, ST2 is also present in other cell types. Thus, it should be IL33/ST2 signaling deficiency and not ILC2 deficiency.

Answer 9

 Thank you for your comment. As you indicated, we have corrected the ILC2 deficiency to IL33/ST2 signaling deficiency.

Discussion(page12)

IL33/ST2 signaling deficiency enlarged atherosclerotic foci, increased inflammatory gene expression levels, and significantly increased saturated fatty acid concentrations within the atherosclerotic foci.

Comment 10

 * In Line 387-388, the authors mention that the increased blood pressure (despite not reaching hypertensive levels) was due to arterial stiffness. I believe this is highly speculative because there was no data on pulse wave velocity.

Answer 10

 Thank you for your comment. As you pointed out, neither ApoEKO nor ApoEST2DKO mice reached hypertensive levels, and pulse wave velocity was not measured, so the worsening of arterial stiffness and the increase in blood pressure are speculative. We have changed the text as follows.

Discussion (Page 13)

Although not reaching hypertensive levels, the reason for the increased systolic blood pressure in the ApoEST2DKO mice group compared to the ApoEKO mice group could be worsening arterial stiffness.

Comment 11

    * Line 3923-395 is also highly speculative and not supported by data. In Fig 4N, O and R, ST2 KO did change the expression of inflammatory genes.

Answer 11

    Thank you for your comment. I have deleted the part you pointed out and replaced it with the following.

Discussion(page13)

In our study, ApoEST2KO mice had altered expression of inflammatory genes(CD36,Il6,Tnfa).

Comment 12

   * Overall, the discussion needs more integration of the results. The way it is written at its current form reads more of a literature review.

Answer 12

      Thank you for your comment. Results have been added to the discussion.

Discusion(page13)

In our study, we observed increased M1/M2 macrophage ratio, increased ILC1 and de-creased ILC2 in both arteries and eWAT in mice lacking ApoEST2.

In the ApoEST2DKO mice in this study, arterial saturated fatty acid concentrations (myristic acid, palmitic acid, and stearic acid) were elevated.

Comment 13

* Can the authors provide the sample size in the figure legends?

Answer 13

      Thank you for your valuable comment. As you indicated, we have included sample sizes in figure.

Thank you for giving us the opportunity to strengthen our manuscript with your valuable comments. We have worked hard to incorporate your feedback and hope that these revisions persuade you to accept our submission.

 We hope the revised version is now suitable for publication and look forward to hearing from you.

Yours faithfully,

Masahide Hamaguchi, MD, PhD

Department of Endocrinology and Metabolism, Kyoto Prefectural University of Medicine, Graduate School of Medical Science

Address: 465 Kajii-cho, Kawaramachi-Hirokoji, Kamigyo-ku, Kyoto 602-8566, Japan

Fax: +81-75-252-3721, Tel: +81-75-251-5506

E-mail: mhama@koto.kpu-m.ac.jp

Reviewer 3 Report

Comments and Suggestions for Authors

The manuscript showed the alleviating effect of IL-33 on atherosclerosis by reducing the accumulation of saturated fatty acids in the aorta, demonstrated potential therapy for atherosclerosis patients. While the data in the manuscript are comprehensive, I have the following minor concerns:

1. The statistical methods in Fig 5 need adjustment. Since it involves comparing data from more than two groups, one-way ANOVA should be used for statistical analysis, followed by appropriate post-hoc tests. Using a t-test in these experiments is inappropriate.

2. Secondly, the manuscript lacks analysis of experimental limitations and plans for future experiments, such as further exploration of pathways in IL-33 promoting metabolism from saturated to unsaturated fatty acid, or supplementation of IL-33 in vivo to reverse atherosclerosis. Please add this part to the Discussion section.

3. Subtitles in Line 278 are not correctly wrapped.

Author Response

Prof. Dr. Maria Luz Fernandez

Prof. Dr. Liuis Serra-Majem

Editor-in-Chief

Nutrients MDPI

                          14 Apr 2024

Dear Editor,

Ref.: Manuscript ID: nutrients-2930866

 Thank you for your kind letter concerning our manuscript.

Enclosed please find our revised manuscript entitled “IL-33 reduces saturated fatty acid accumulation in mouse atherosclerotic foci”, manuscript ID of which is nutrients-2930866.

 At first, we would like to thank reviewers for their constructive comments on our manuscript. Based on the reviewers’ comments, we have revised our manuscript. The responses to reviewers’ comments are described below. Your kind consideration of this paper would be greatly appreciated.

Response to Reviewer 3

Major comments

Comment 1

 *  The statistical methods in Fig 5 need adjustment. Since it involves comparing data from more than two groups, one-way ANOVA should be used for statistical analysis, followed by appropriate post-hoc tests. Using a t-test in these experiments is inappropriate.

Answer 1

 Thank you very much for your comment. For Figure 5, the following text was added because a 1-way ANOVA multiple comparison test was used in the analysis.

Materials and Methods【2.12 Statistical Analyses (Page 5)】

Comparisons of three or more groups were analyzed using 1-way ANOVA multiple comparison tests.

Comment 2

 * Secondly, the manuscript lacks analysis of experimental limitations and plans for future experiments, such as further exploration of pathways in IL-33 promoting metabolism from saturated to unsaturated fatty acid, or supplementation of IL-33 in vivo to reverse atherosclerosis. Please add this part to the Discussion section.

Answer 2

 Thank you for your valuable comment. The following is included in the discussion.

Discussion section (Page 14)

There are limitations to this study. It is possible that ST2 signaling in endothelial cells has some effect on NO signaling, but we have not been able to measure NO in our study. Further pathways by which IL-33 promotes the metabolism of saturated fatty acids to unsaturated fatty acids have not been elucidated. In vivo supplementation of IL-33 to reverse atherosclerosis is a topic for future research to clarify the validity of this study.

Comment 3

 * Subtitles in Line 278 are not correctly wrapped.

Answer 3

    Thank you for your comment. I have redacted the part you pointed out.

3.4. Aortic Inflammation and Anti-inflammatory Cell Populations (Page 8)

 Thank you for giving us the opportunity to strengthen our manuscript with your valuable comments. We have worked hard to incorporate your feedback and hope that these revisions persuade you to accept our submission.

 We hope the revised version is now suitable for publication and look forward to hearing from you.

Yours faithfully,

Masahide Hamaguchi, MD, PhD

Department of Endocrinology and Metabolism, Kyoto Prefectural University of Medicine, Graduate School of Medical Science

Address: 465 Kajii-cho, Kawaramachi-Hirokoji, Kamigyo-ku, Kyoto 602-8566, Japan

Fax: +81-75-252-3721, Tel: +81-75-251-5506

E-mail: mhama@koto.kpu-m.ac.jp